# LEARNING TO REMEMBER RARE EVENTS

**Łukasz Kaiser**[*]
Google Brain
lukaszkaiser@google.com

**Ofir Nachum**[*†]
Google Brain
ofirnachum@google.com

**Aurko Roy**[‡]
Georgia Tech
aurko@gatech.edu

**Samy Bengio**
Google Brain
bengio@google.com

## ABSTRACT

Despite recent advances, memory-augmented deep neural networks are still limited when it comes to life-long and one-shot learning, especially in remembering rare events. We present a large-scale life-long memory module for use in deep learning. The module exploits fast nearest-neighbor algorithms for efficiency and thus scales to large memory sizes. Except for the nearest-neighbor query, the module is fully differentiable and trained end-to-end with no extra supervision. It operates in a life-long manner, i.e., without the need to reset it during training.

Our memory module can be easily added to any part of a supervised neural network. To show its versatility we add it to a number of networks, from simple convolutional ones tested on image classification to deep sequence-to-sequence and recurrent-convolutional models. In all cases, the enhanced network gains the ability to remember and do life-long one-shot learning. Our module remembers training examples shown many thousands of steps in the past and it can successfully generalize from them. We set new state-of-the-art for one-shot learning on the Omniglot dataset and demonstrate, for the first time, life-long one-shot learning in recurrent neural networks on a large-scale machine translation task.

## 1 INTRODUCTION

Machine learning systems have been successful in many domains, from computer vision (Krizhevsky et al., 2012) to speech recognition (Hinton et al., 2012) and machine translation (Sutskever et al., 2014; Bahdanau et al., 2014; Cho et al., 2014). Neural machine translation (NMT) is so successful that for some language pairs it approaches, *on average*, the quality of human translators (Wu et al., 2016). The words *on average* are crucial though. When a sentence resembles one from the abundant training data, the translation will be accurate. However, when encountering a rare word such as *Dostoevsky* (in German, *Dostojewski*), many models will fail. The correct German translation of *Dostoevsky* does not appear enough times in the training data for the model to sufficiently learn its translation.

While more example sentences concerning the famous Russian author might eventually be added to the training data, there are many other rare words or rare events of other kinds. This illustrates a general problem with current deep learning models: it is necessary to extend the training data and re-train them to handle such rare or new events. Humans, on the other hand, learn in a life-long fashion, often from single examples.

We present a life-long memory module that enables one-shot learning in a variety of neural networks. Our memory module consists of key-value pairs. Keys are activations of a chosen layer of a neural network, and values are the ground-truth targets for the given example. This way, as the network is trained, its memory increases and becomes more useful. Eventually it can give predictions that

---

[*]First two authors contributed equally.
[†]Work done as a member of the Google Brain Residency program (g.co/brainresidency).
[‡]Work done during internship at Google Brain.

leverage on knowledge from past data with similar activations. Given a new example, the network writes it to memory and is able to use it afterwards, even if the example was presented just once.

There are many advantages of having a long-term memory. One-shot learning is a desirable property in its own right, and some tasks, as we will show below, are simply not solvable without it. Even real-world tasks where we have large training sets, such as translation, can benefit from long-term memory. Finally, since the memory can be traced back to training examples, it might help explain the decisions that the model is making and thus improve understandability of the model.

It is not immediately clear how to measure the performance of a life-long one-shot learning model, since most deep learning evaluations focus on the average performance and do not have a one-shot component. We therefore evaluate in a few ways, to show that our memory module indeed works:

(1) We evaluate on the well-known one-shot learning task Omniglot, which is the only dataset with explicit one-shot learning evaluation. This dataset is small and does not benefit from life-long learning capability of our module, but we still exceed the best previous results and set new state-of-the-art.

(2) We devise a synthetic task that requires life-long one-shot learning. On this task, standard models fare poorly while our model can solve it well, demonstrating its strengths.

(3) Finally, we train an English-German translation model that has our life-long one-shot learning module. It retains very good performance on average and is also capable of one-shot learning. On the qualitative side, we find that it can translate rarely-occurring words like Dostoevsky. On the quantitative side, we see that the BLEU score for the generated translations can be significantly increased by showing it related translations before evaluating.

## 2   MEMORY MODULE

Our memory consists of a matrix $K$ of memory keys, a vector $V$ of memory values, and an additional vector $A$ that tracks the age of items stored in memory. Keys can be arbitrary vectors of size `key-size`, and we assume that the memory values are single integers representing a class or token ID. We define a memory of size `memory-size` as a triple:

$$\mathcal{M} = (K_{\texttt{memory-size}\times\texttt{key-size}}, V_{\texttt{memory-size}}, A_{\texttt{memory-size}}).$$

A memory query is a vector of size `key-size` which we assume to be normalized, i.e., $\|q\| = 1$. Given a query $q$, we define the nearest neighbor of $q$ in $\mathcal{M}$ as any of the keys that maximize the dot product with $q$:

$$\text{NN}(q, \mathcal{M}) = \text{argmax}_i \, q \cdot K[i].$$

Since the keys are normalized, the above notion corresponds to the nearest neighbor with respect to cosine similarity. We will also use the natural extension of it to $k$ nearest neighbors, which we denote $\text{NN}_k(q, \mathcal{M})$. In our experiments we always used the set of $k = 256$ nearest neighbors.

When given a query $q$, the memory $\mathcal{M} = (K, V, A)$ will compute $k$ nearest neighbors (sorted by decreasing cosine similarity):

$$(n_1, \ldots, n_k) = \text{NN}_k(q, \mathcal{M})$$

and return, as the main result, the value $V[n_1]$. Additionally, we will compute the cosine similarities $d_i = q \cdot K[n_i]$ and return $\text{softmax}(d_1 \cdot t, \ldots, d_k \cdot t)$. The parameter $t$ denotes the inverse of softmax temperature and we set it to $t = 40$ in our experiments. In models where the memory output is again embedded into a dense vector, we multiply the embedded output by the corresponding softmax component so as to provide a signal about confidence of the memory.

The forward computation of the memory module is thus very simple, the only interesting part being how to compute nearest neighbors efficiently, which we discuss below. But we must also answer the question how the memory is trained.

**Memory Loss.**   Assume now that in addition to a query $q$ we are also given the correct desired (supervised) value $v$. In the case of classification, this $v$ would be the class label. In a sequence-to-sequence task, $v$ would be the desired output token of the current time step. After computing the $k$ nearest neighbors $(n_1, \ldots, n_k)$ as above, let $p$ be the smallest index such that $V[n_p] = v$ and

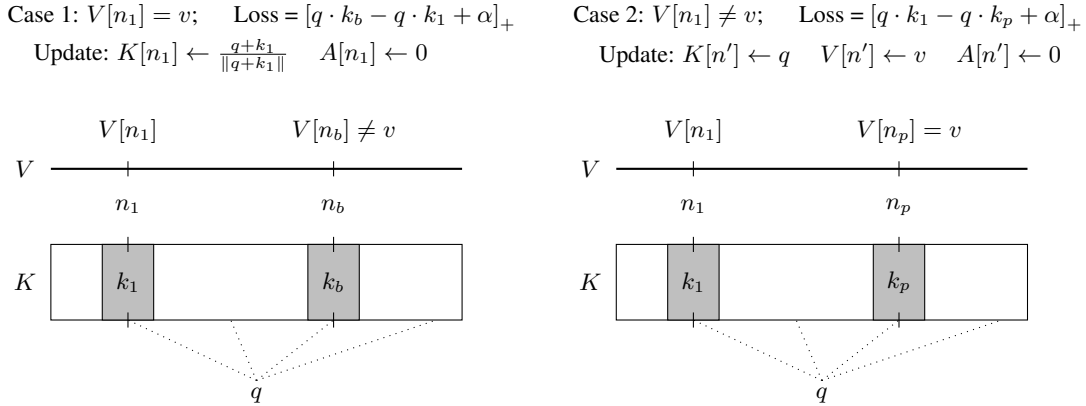

Case 1: $V[n_1] = v;$     Loss = $[q \cdot k_b - q \cdot k_1 + \alpha]_+$     Case 2: $V[n_1] \neq v;$     Loss = $[q \cdot k_1 - q \cdot k_p + \alpha]_+$

Update: $K[n_1] \leftarrow \frac{q+k_1}{\|q+k_1\|}$     $A[n_1] \leftarrow 0$     Update: $K[n'] \leftarrow q$     $V[n'] \leftarrow v$     $A[n'] \leftarrow 0$

Figure 1: The operation of the memory module on a query $q$ with correct value $v$; see text for details.

$b$ the smallest index such that $V[n_b] \neq v$. We call $n_p$ the *positive neighbor* and $n_b$ the *negative neighbor*. When no positive neighbor is among the top-$k$, we pick any vector from memory with value $v$ instead of $K[n_p]$. We define the memory loss as:

$$\text{loss}(q, v, \mathcal{M}) = [q \cdot K[n_b] - q \cdot K[n_p] + \alpha]_+ .$$

Recall that both $q$ and the keys in memory are normalized, so the products in the above loss term correspond to cosine similarities between $q$, the positive key, and the negative key. Since cosine similarity is maximal for equal terms, we want to maximize the similarity to the positive key and minimize the similarity to the negative one. But once they are far enough apart (by the margin $\alpha$, 0.1 in all our experiments), we do not propagate any loss. This definition and reasoning behind it are almost identical to the one in Schroff et al. (2015) and similar to many other distance metric learning works (Weinberger & Saul, 2009; Weston et al., 2011).

**Memory Update.**    In addition to computing the loss, we will also update the memory $\mathcal{M}$ to account for the fact that the newly presented query $q$ corresponds to $v$. The update is done in a different way depending on whether the main value returned by the memory module already is the correct value $v$ or not. As before, let $n_1 = \text{NN}(q, \mathcal{M})$ be the nearest neighbor to $q$.

If the memory already returns the correct value, i.e., if $V[n_1] = v$, then we only update the key for $n_1$ by taking the average of the current key and $q$ and normalizing it:

$$K[n_1] \leftarrow \frac{q + K[n_1]}{\|q + K[n_1]\|}.$$

When doing this, we also re-set the age: $A[n_1] \leftarrow 0$.

Otherwise, when $V[n_1] \neq v$, we find a new place in the memory and write the pair $(q, v)$ there. Which place should we choose? We find memory items with maximum age, and write to one of those (randomly chosen). More formally, we pick $n' = \text{argmax}_i A[i] + r_i$ where $|r_i| \ll |\mathcal{M}|$ is a random number that introduces some randomness in the choice so as to avoid race conditions in asynchronous multi-replica training. We then set:

$$K[n'] \leftarrow q, \quad V[n'] \leftarrow v, \quad A[n'] \leftarrow 0.$$

With every memory update we also increment the age of all non-updated indices by 1. The full operation of the memory module is depicted in Figure 1.

**Efficient nearest neighbor computation.**    The most expensive operation in our memory module is the computation of $k$ nearest neighbors. This can be done exactly or in an approximate way.

In the exact mode, to calculate the nearest neighbors in $K$ to a mini-batch of queries $Q = (q_1, \dots, q_b)$, we perform a single matrix multiplication: $Q \times K^T$. This multiplies the `batch-size`

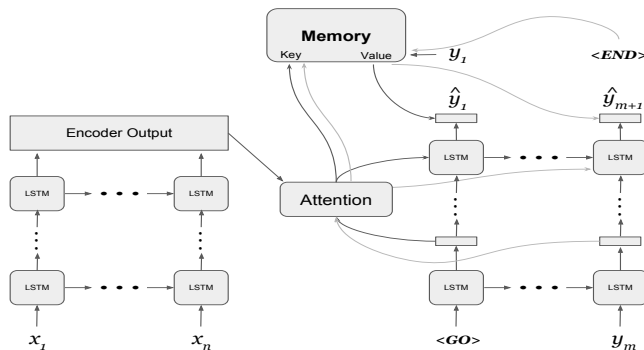

Figure 2: The GNMT model with added memory module. On each decoding step $t$, the result of the attention $a_t$ is used to query the memory. The resulting value is combined with the output of the final LSTM layer to produce the predicted logits $\hat{y}_t$. See text for further details.

$\times$ `key-size` matrix $Q$ by the `key-size` $\times$ `memory-size` matrix $K^T$, and the result is the `batch-size` $\times$ `memory-size` matrix of all distances, from which we can choose the top-$k$. This procedure is linear in `memory-size`, so it can be expensive for very large memory sizes. But matrix multiplication is very heavily optimized, so in our experiments on GPUs we find that this operation is not a bottleneck for memory sizes up to half a million.

If the exact mode is too slow, the $k$ nearest neighbors can be computed approximately using locality sensitive hashing (LSH). LSH is a hashing scheme so that near neighbors get similar hashes (Indyk & Motwani, 1998; Andoni & Indyk, 2006). For cosine similarity, the computation of an LSH is very simple. We pick a number of random normalized hash vectors $h_1, \ldots, h_l$. The hash of a query $q$ is a sequence of $l$ bits, $b_1, \ldots, b_l$, such that $b_i = 1$ if, and only if, $q \cdot h_i > 0$. It turns out that near neighbors will, with high probability, have a large number of identical bits in their hash. To compute the nearest neighbors it is therefore sufficient to only look into parts of the memory with similar hashes. This makes the nearest neighbor computation work in approximately constant time – we only need to multiply the query by the hash vectors, and then only use the nearest buckets.

## 2.1 USING THE MEMORY MODULE

The memory module presented above can be added to any classification network. There are two main choices: which layer to use to generate queries, and how to use the output of the module.

In the simplest case, we use the final layer of a network as query and the output of the module is directly used for classification. This simplest case is similar to matching networks (Oriol Vinyals, 2016b) and our memory module yields good results already in this setting (see below).

Instead of using the output of the module directly, it is possible to embed it again into a dense representation and mix it with other predictions made by the network. To study this setting, we add the memory module to sequence-to-sequence recurrent neural networks. As described in detail below, a query to memory is made in every step of the decoder network. Memory output is embedded again into a dense representation and combined with inputs from other layers of the network.

**Convolutional Network with Memory.** To test our memory module in a simple setting, we first add it to a basic convolutional network network for image classification. Our network consists of two convolutional layers with ReLU non-linearity, followed by a max-pooling layer, another two convolutional-ReLU layers, another max-pooling, and two fully connected layers. All convolutions use $3 \times 3$ filters with $64$ channels in the first pair, and $128$ in the second. The fully connected layers have dimension $256$ and dropout applied between them. The output of the final layer is used as query to our memory module and the nearest neighbor returned by the memory is used as the final network prediction. Even this basic architecture yields good results in one-shot learning, as discussed below.

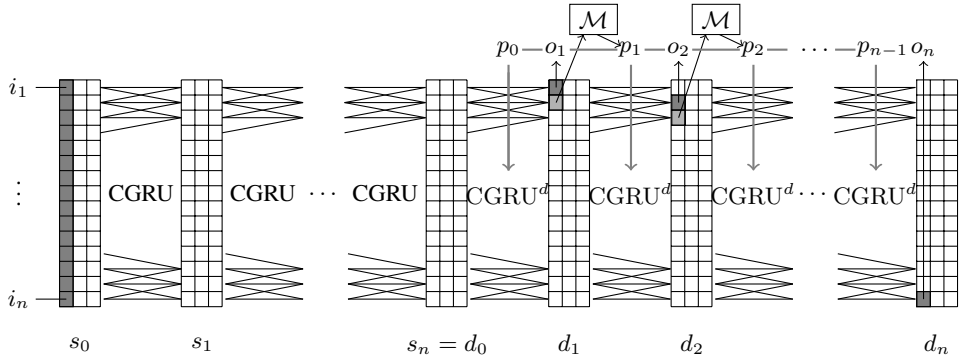

Figure 3: Extended Neural GPU with memory module. Memory query is read from the position one below the current output logit, and the embedded memory value is put at the same position of the output tape $p$. The network learns to use these values to produce the output in the next step.

**Sequence-to-sequence with Memory.** For large-scale experiments, we add the memory module into a large sequence-to-sequence model. Such sequence-to-sequence recurrent neural networks (RNNs) with long short-term memory (LSTM) cells (Hochreiter & Schmidhuber, 1997) have proven especially successful at natural language processing (NLP) tasks, including machine translation (Sutskever et al., 2014; Bahdanau et al., 2014; Cho et al., 2014). We add the memory module to the Google Neural Machine Translation (GNMT) model (Wu et al., 2016). This model consists of an encoder RNN, which creates a representation of the source language sentence, and a decoder RNN that outputs the target language sentence. We left the encoder RNN unmodified. In the decoder RNN, we use the vector retrieved by the attention mechanism as query to the memory module. In the GNMT model, the attention vector is used in all LSTM layers beyond the second one, so the computation of the other layers and the memory can happen in parallel. Before the final softmax layer, we combine the embedded memory output with the output of the final LSTM layer using an additional linear layer, as depicted in Figure 2.

**Extended Neural GPU with Memory.** To test versatility of our memory module, we also add it to the Extended Neural GPU, a convolutional-recurrent model introduced by Kaiser & Bengio (2016). The Extended Neural GPU is a sequence-to-sequence model too, but its decoder is convolutional and the size of its state changes depending on the size of the input. Again, we leave the encoder part of the model intact, and extend the decoder part by a memory query. This time, we use the position one step ahead to query memory, and we put the embedded result to the output tape, as shown in Figure 3. Note that in this model the result of the memory will be processed by two recurrent-convolutional cells before the corresponding output is produced. The fact that this model still does one-shot learning confirms that the output of our memory module can be used deep inside a network, not just near the output layer.

## 3 RELATED WORK

**Memory in Neural Networks.** Augmenting neural networks with memory has been heavily studied recently. Many of these approaches design a memory component that is intended as a generalization of the memory in standard recurrent neural networks. In recurrent networks, the state passed from one time step to the next can be interpreted as the network's memory representation of the current example. Moving away from this fixed-length vector representation of memory to a larger and more versatile form is at the core of these methods.

Augmenting recurrent neural networks with attention (Bahdanau et al., 2014) can be interpreted as creating a large memory component that allows content-based addressing. More generally, Graves et al. (2014) augmented a recurrent neural network with a computing-inspired memory component that can be addressed via both content- and address-based queries. Sukhbaatar et al. (2015) present a similar augmentation and show the importance of allowing multiple reads and writes to memory between inputs. These approaches excel at tasks where it is necessary to store large parts of a se-

quential input in a representation that can later be precisely queried. Such tasks include algorithmic sequence manipulation tasks, natural language modelling, and question-answering tasks.

The success of these approaches hinges on making the memory component fully differentiable and backpropagating signal through every access of memory. In this setting, computational requirements necessitate that the memory be small. Some attempts have been made at making hard access queries to memory (Zaremba & Sutskever, 2015; Xu et al., 2015), but it was usually challenging to match the soft version. Recently, more successful training for hard queries was reported (Gülçehre et al., 2016) that makes use of a curriculum strategy that mixes soft and hard queries at training time. Our approach applies hard access as well, but we encourage the model to make good queries via a special memory loss.

Modifications to allow for large-scale memory in neural networks have been proposed. The original implementation of memory networks (Weston et al., 2014) and later work on scaling it (Bordes et al., 2015; Chandar et al., 2016) used memory with size in the millions. The cost of doing so is that the memory must be fixed prior to training. Moreover, since during the beginning of training the model is unlikely to query the memory correctly, strong supervision is used to encourage the model to query memory locations that are useful. These hints are either given as additional supervising information by the task or determined heuristically as in Hill et al. (2015).

All the work discussed so far has either used a memory that is fixed before training or used a memory that is not persistent between different examples. For one-shot and lifelong learning, a memory must necessarily be both volatile during training and persistent between examples. To bridge this gap, Santoro et al. (2016) propose to partition training into distinct episodes consisting of a sequence of labelled examples $\{(x_i, y_i)\}_{i=1}^n$. A network augmented with a fully-differentiable memory is trained to predict $y_i$ given the previous sequence $(x_1, y_1, \ldots, x_{i-1})$. This way, the model learns to store important examples with their corresponding labels in memory and later re-use this information to correctly classify new examples. This model successfully exhibits one-shot learning on Omniglot.

However, this approach again requires fully-differentiable memory access and thus limits the size of the memory as well as the length of an episode. This restriction has recently been alleviated by Rae et al. (2016). Their model can utilize large memories, but unlike our work does not have an explicit cost to guide the formation of memory keys.

For classification tasks like Omniglot, it is easy to construct short episodes so that they include a few examples from each of several classes. However, this becomes harder as the output becomes richer. For example, in the difficult sequence-to-sequence tasks which we consider, it is hard to determine which examples would be helpful for correctly predicting others *a priori*, and so constructing short episodes each containing examples that are similar and act as hints to each other is intractable.

**One-shot Learning.** While the recent work of Santoro et al. (2016) succeeded in bridging the gap between memory-based models and one-shot learning, the field of one-shot learning has seen a variety of different approaches over time.

Early work utilized Bayesian methods to model data generatively (Fei-Fei et al., 2006; Lake et al., 2011). The paper that introduced the Omniglot dataset (Lake et al., 2011) approached the task with a generative model for strokes. This way, given a single character image, the probability of a different image being of the same character may be approximated via standard techniques. One early neural network approach to one-shot learning was given by Siamese networks (Koch, 2015). When our approach is applied to the Omniglot image classification dataset, the resulting training algorithm is actually similar to that of Siamese networks. The only difference is in the loss function: Siamese networks utilize a cross-entropy loss whereas our method uses a margin triplet loss.

A more sophisticated neural network approach is given by Vinyals et al. (2016). The strengths of this approach are (1) the model architecture utilizes recent advances in attention-augmented neural networks for set-to-set learning (Oriol Vinyals, 2016a), and (2) the training algorithm is designed to exactly match the testing phase (given $k$ distinct images and an additional image, the model must predict which of the $k$ images is of the same class as the additional image). This approach may also be considered as a generalization of previous work on metric learning.

Table 1: Results on the Omniglot dataset. Although our model uses only a simple convolutional neural network, the addition of our memory module allows it to approach much more complex models on 1-shot and multi-shot learning tasks.

| Model | 5-way 1-shot | 5-way 5-shot | 20-way 1-shot | 20-way 5-shot |
|---|---|---|---|---|
| Pixels Nearest Neighbor | 41.7% | 63.2% | 26.7% | 42.6% |
| MANN (no convolutions) | 82.8% | 94.9% | – | – |
| Convolutional Siamese Net | 96.7% | 98.4% | 88.0% | 96.5% |
| Matching Network | 98.1% | 98.9% | 93.8% | 98.5% |
| ConvNet with Memory Module | 98.4% | 99.6% | 95.0% | 98.6% |

## 4 EXPERIMENTS

We perform experiments using all three architectures described above. We experiment both on real-world data and on synthetic tasks that give us some insight into the performance and limitations of the memory module. In all our experiments we use the Adam optimizer (Kingma & Ba, 2014) and the parameters for the memory module remain unchanged ($k = 256, \alpha = 0.1$). Good performance with a single set of parameters shows the versatility of our memory module. The source code for the memory module, together with our settings for Omniglot, is available on github[1].

**Omniglot.** The Omniglot dataset (Lake et al., 2011) consists of 1623 characters from 50 different alphabets, each hand-drawn by 20 different people. The large number of classes (characters) with relatively few data per class (20), makes this an ideal data set for testing one-shot classification. In the $N$-way Omniglot task setup we pick $N$ unseen character classes, independent of alphabet. We provide the model with one drawing of each character and measure its accuracy the $K$-th time it sees the character class. Our setup is identical to Oriol Vinyals (2016b), so we also augmented the data set with random rotations by multiples of 90 degrees and use 1200 characters for training, and the remaining character classes for evaluation. We present the results from Oriol Vinyals (2016b) and ours in Table 1. Even with a simpler network without batch normalization, we get similar results.

**Synthetic task.** To better understand the memory module operation and to test what it can remember, we devise a synthetic task and train the Extended Neural GPU with and without memory (we use a small Extended Neural GPU with 32 channels and memory of size half a million).

To create training and test data for our synthetic task, we use symbols from the set $S = \{2, \ldots, 16000\}$ and first fix a random function $f : S \to S$. The function $f$ is chosen at random, but fixed and the same for all training and testing examples (we used 40K training examples).

In our synthetic task, the input is a sequence consisting of As and Bs with one continuous substring of 7 digits from the set $0, 1, 2, 3$. The substring is interpreted as a number written in base-4, e.g., $1982 = 132332_4$, so the string 132332 would be interpreted as 1982. The corresponding output is created by copying all As and Bs, but mapping the number through the random function $f$. For instance, assuming $f(1982) = 3726$, the output corresponding to 132332 would be 322032 as $3726 = 322032_4$. Here is an example of an input-output pair:

| Input | A | 0 | 1 | 3 | 2 | 3 | 3 | 2 | B | A | B | A | B |
|---|---|---|---|---|---|---|---|---|---|---|---|---|---|
| Output | A | 0 | 3 | 2 | 2 | 0 | 3 | 2 | B | A | B | A | B |

This task clearly requires memory to store the fixed random function. Since there are 16K elements to learn, it is hard to memorize, and each single instance occurs quite rarely. The raw Extended Neural GPU (or any other sequence-to-sequence model) are limited by their size. With long training, the small model can memorize some of the sequences, but it is only a small fraction.

Additionally, there is no direct indication in the data what part of the input should trigger the production of each output symbol. For example, to produce the first 3 output in the above example, the

---

[1] `https://github.com/tensorflow/models/tree/master/learning_to_remember_rare_events`

Table 2: Results on the synthetic task. We report the percentage of fully correct sequences from the test set, which contains 10000 random examples. See text for details.

| Model | Accuracy |
|---|---|
| Hamming Nearest Neighbor | 0.1% |
| Baseline Sequence-to-Sequence with Attention | 0.9% |
| Baseline Extended Neural GPU | 12.2% |
| Sequence-to-Sequence with Attention and Memory | 35.2% |
| Extended Neural GPU with Memory Module | 71.3% |

Table 3: Results on the WMT En-De task. As described in the text, we split the test set in two (odd lines and even lines) to evaluate the model on one-shot learning. Given the even test set, the model can perform better on the odd test set. We also see a dramatic improvement when the model is provided with the whole test set, validating that the memory module is working as intended.

| Model | Full Test | Odd Test |
|---|---|---|
| GNMT | 23.25 | 23.17 |
| GNMT with Memory Module | 23.29 | 23.16 |
| GNMT with Memory Module and Even Test context | – | 23.60 |
| GNMT with Memory Module and Whole Test context | 31.11* | – |

memory key needs to encode all base-4 symbols from the input. Not just one or two aligned symbols, but a number of them. Moreover, it should not encode more symbols or it will not generalize to the test set. Similarly, a basic nearest neighbor classifier fails on this task. We use sequences of length up to 40 during training, but there are only 7 relevant symbols. The simple nearest neighbor by Hamming distance will most probably select some sequence with similar prefix or suffix of As and Bs, and not the one with the corresponding base-4 part. We also trained a large sequence-to-sequence model with attention on this task (a 2-layer LSTM model with 256 units in each layer). This model can memorize the whole training set, but it suffers from a similar problem as the Hamming nearest neighbor – it almost doesn't generalize, its accuracy on the test set is only about 1%. The same model with a memory module generalizes much better, reaching over 30% accuracy. The Extended Neural GPU with our memory module yields even better results, see Table 2.

**Translation.** To evaluate the memory module in a large-scale setting we use the GNMT model (Wu et al., 2016) extended with our memory module on the WMT14 English-to-German translation task. We evaluate the model both qualitatively and quantitatively.

On the qualitative side, we note that our memory-augmented model can successfully translate rare words like Dostoevsky, unlike the baseline model which predicts an identity-mapped *Dostoevsky* for the German translation of Dostoevsky.

On the quantitative side, we use the WMT test set. We find that in terms of BLEU score, an aggregate measure, the memory-augmented GNMT is on par with the baseline GNMT, see Table 3.

To evaluate our memory-augmented model for one-shot capabilities we split the test set in two. We take the even lines of the test set (index starting at 0) as a context set and the odd lines of the test set as the one-shot evaluation set. While showing the context set to the model, no additional training occurs, only memory updates are allowed. So the weights of the model do not change, but the memory does. Since the sentences in the test set are highly-correlated to each other (they come from paragraphs with preserved order), we expect that if we allow a one-shot capable model to use the context set to update its memory and then evaluate it on the other half of the test set, its accuracy will increase. For our GNMT with memory model, we passed the context set through the memory update operations 3 times. As seen in Table 3, the context set indeed helps when evaluating on the odd lines, increasing the BLEU score by almost 0.5. As further indication that our memory module works properly, we also evaluate the model after showing the whole test set as a context set. Note that this is essentially an oracle: the memory module gets to see all the correct answers, we do this only to test and debug. As expected, this increases BLEU score dramatically, by over 8 points.

## 5 DISCUSSION

We presented a long-term memory module that can be used for life-long learning. It is versatile, so it can be added to different deep learning models and at different layers to give the networks one-shot learning capability. Several parts of the presented memory module could be tuned and studied in more detail. The update rule that averages the query with the correct key could be parametrized. Instead of returning only the single nearest neighbor we could also return a number of them to be processed by other layers of the network. We leave these questions for future research.

The main issue we encountered, though, is that evaluating one-shot learning is difficult, as standard metrics do not focus on this scenario. In this work, we adapted the standard metrics to investigate our approach. For example, in the translation task we used half of the test set as context for the other half, and we still report the standard BLEU score. This allows us to show that our module works, but it is only a temporary solution. Better metrics are needed to accelerate progress of one-shot and life-long learning. Thus, we consider the present work as just a first step on the way to making deep models learn to remember rare events through their lifetime.

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
