# Peer review of "Learning to Remember Rare Events"

_ICLR 2017 — accepted_

[Author Response · Lukasz Kaiser · 12 Dec 2016]
**New Revision**

We are very grateful for the reviewers' questions, and we uploaded a new revision that clarifies the paper to answer them.

In addition to the answers, we updated the definition of the memory module for the cases when no positive neighbour is found in the top-k. We now get any vector from memory instead of using all 0s, which prevents the loss from jumping in such cases. This improved our results on Omniglot, they are state-of-the-art now.

[Official Review · AnonReviewer3 · rating 7 · confidence 4 · 16 Dec 2016]
**interesting new memory module**

The paper proposes a new memory module to be used as an addition to existing neural network models.

Pros:
* Clearly written and original idea.
* Useful memory module, shows nice improvements.
* Tested on some big tasks.

Cons:
* No comparisons to other memory modules such as associative LSTMs etc.

[Official Review · AnonReviewer1 · rating 8 · confidence 3 · 19 Dec 2016]
**No Title**

A new memory module based on k-NN is presented.
The paper is very well written and the results are convincing. 

Omniglot is a good sanity test and the performance is surprisingly good.
The artificial task shows us that the authors claims hold and highlight the need for better benchmarks in this domain.
And the translation task eventually makes a very strong point on practical usefulness of the proposed model.

I am not a specialist in memory networks so I trust the authors to double-check if all relevant references have been included (another reviewer mentioned associative LSTM). But besides that I think this is a very nice and useful paper. I hope the authors will publish their code.

[Official Review · AnonReviewer2 · rating 6 · confidence 5 · 20 Dec 2016]

This paper proposes a new memory module for large scale life-long and one-shot learning. The module is general enough that the authors apply the module to several neural network architectures and show improvements in performance.

Using k-nearest neighbors for memory access is not completely new. This has been recently explored in Rae et al., 2016 and Chandar et al., 2016. K-nearest neighbors based memory for one-shot learning has also been explored in [R1]. This paper provides experimental evidence that such an approach can be applied to a variety of architectures.

Authors have addressed all my pre-review questions and I am ok with their response.

Are the authors willing to release the source code to reproduce the results? At least for omniglot experiments and synthetic task experiments?

References:

[R1] Charles Blundell, Benigno Uria, Alexander Pritzel, Yazhe Li, Avraham Ruderman, Joel Z. Leibo, Jack Rae, Daan Wierstra, Demis Hassabis: Model-Free Episodic Control. CoRR abs/1606.04460 (2016)

[Public Comment · (anonymous) · 17 Jan 2017]
**Question about update memory during training**

what does "introduce some randomness in the choice so as to avoid race conditions in asynchronous multi-replica training" mean specifically?
Is there any reference paper to let the readers to get better understanding of it? Thank you very much.

[Final Decision · Program Chairs · 06 Feb 2017]
**ICLR committee final decision**

The primary contribution of this paper is showing that k-nearest-neighbor method based memory can be usefully incorporated in a variety of architectures and supervised learning tasks. The presentation is clear, and results are good. I like the synthetic task and analysis. For the Omniglot task, running and reporting results on the original splits used by Lake would be good, as the splits used in matching nets are considerably easier and result in ceiling effects.